# Convergence and transdisciplinary teaching in quantitative biology

Robert Mayes[1] , Joseph Dauer[2] and David Owens[1]

[1]Georgia Southern University, Statesboro, GA, United States; [2]University of Nebraska—Lincoln, Lincoln, NE, USA

## Perspective

convergence; transdisciplinary; quantitative biology.

**Corresponding author:**
Joseph Dauer;
Email: joseph.dauer@unl.edu

### Abstract

The United States National Science and Technology Council has made a call for improving STEM (Science, Technology, Engineering, and Mathematics) education at the convergence of science, technology, engineering, and mathematics. The National Science Foundation (NSF) views *convergence* as the merging of ideas, approaches, and technologies from widely diverse fields of knowledge to stimulate innovation and discovery. Teaching convergency requires moving to the transdisciplinary level of integration where there is deep integration of skills, disciplines, and knowledge to solve a challenging real-world problem. Here we present a summary on convergence and transdisciplinary teaching. We then provide examples of convergence and transdisciplinary teaching in plant biology, and conclude by discussing limitations to contemporary conceptions of convergency and transdisciplinary STEM.

## 1. Introduction

The National Science Foundation (NSF) has included convergence in the "NSF 10 Big Ideas" for the future of STEM (Science, Technology, Engineering, and Mathematics), stating that the grand challenges of tomorrow, such as human health (Dzau & Balatbat, 2018; Sharp & Hockfield, 2017), the food, energy, and water nexus, and exploring the universe, must be solved at the convergence of STEM fields (Smith & Baru, 2020). NSF views *convergence* as the merging of ideas, approaches, and technologies from widely diverse fields of knowledge to stimulate innovation and discovery (National Research Council, n.d.). This effort is supported through the NSF Convergence Accelerator program which addresses national-scale societal challenges through research at the convergence of STEM fields. The call for this program identifies convergence approaches including human-centered design, user discovery, and team science and integration of multidisciplinary research.

How do we develop the next generation of STEM professionals so they can work at the convergence of STEM fields? The Federal STEM Education Strategic Plan (Committee on STEM Education, 2018) identified transdisciplinary teaching as a key pedagogical strategy for teaching STEM in the areas of convergence. *Transdisciplinary* teaching engages students in applying knowledge and skills from two or more disciplines to undertake a real-world problem that shapes the learning experience.

The authors of this article have published several articles around issues of modelling and interdisciplinary STEM, which are related to the concepts of convergence and transdisciplinary teaching. In *Assessing Quantitative Modelling Practices, Metamodelling, and Capability Confidence of Biology Undergraduate Students* (Dauer et al., 2021) the team reported on student ability in quantitative reasoning at the convergence of STEM fields within undergraduate biology. A key finding was that students had confidence in their ability to develop quantitative models for biological phenomena, even while their performance on modeling questions was quite low. *Development of Interdisciplinary STEM Impact Measures of Student Attitudes and Reasoning* (Mayes & Rittschof, 2021) presented on the Real STEM Project, which focused on the development of interdisciplinary STEM reasoning abilities within a real-world context. Interdisciplinary integration in STEM is a level of integration leading up to transdisciplinary integration. The impact of interdisciplinary STEM courses for middle school and high school positively impacted student attitudes towards STEM, but there was no significant improvement in interdisciplinary STEM ability. *Undergraduate Quantitative Biology Impact on Biology Preservice Teachers* (Mayes et al., 2020) discussed the incorporation of quantitative reasoning

into undergraduate biology, specifically to serve the needs of biology preservice teachers. This article had elements of convergence and transdisciplinary teaching in the area of biology.

In this paper, the authors have two goals:

1. Expand beyond interdisciplinary STEM to discuss convergence and transdisciplinary STEM education
2. Focus on plant biology as a subject

The first of these goals reflects the Committee on STEM Education (2018) call for improving STEM education at the convergence of science, technology, engineering, and mathematics. The second goal to focus on plant biology was requested by the journal editor. Transdisciplinary STEM, by definition, must be deliberated within a real-world context and plant biology serves as a rich context for our discussion. We begin by discussing the Federal STEM Education Strategic Plan, then define STEM convergence and transdisciplinary STEM. We finish by providing examples of convergence and transdisciplinary STEM in plant biology.

## 2. Federal STEM education strategic plan

The federal government has established a Federal STEM Education Strategic Plan (Committee on STEM Education, 2018). The National Science Technology Council (NSTC) is directly under the White House Executive Branch and includes the Vice President, the Director of the Office of Science and Technology Policy, and Cabinet Secretaries and Agency Heads with significant science and technology responsibilities. NSTC coordinates science and technology policy across the Federal research and development enterprise to ensure policy decisions and programs align with the President's goals. The NSTC prepares research and development strategies and coordinates them across Federal agencies to achieve national goals. The work of NSTC is organised under committees that oversee subcommittees, working groups, and taskforces focused on science and technology. The Committee on STEM Education (CoSTEM) spearheads the education activities for NSTC. The White House Office of Science and Technology Policy (OSTP) advises the President on topics including the scientific, engineering, and technological aspects of the economy, national security, homeland security, health, foreign relations, the environment, and the technological recovery and use of resources. OSTP leads interagency science and technology policy coordination efforts. The Federal Coordination in STEM Education Subcommittee (FC STEM) has three objectives (Committee on STEM Education, 2018):

1. Help participants in STEM work-based learning programs at federal agencies transition into permanent Federal employees which is critical for the retention of talented trainees and increasing diversity of the Federal STEM workforce
2. Establishment of a single, searchable, user-friendly online resource for finding STEM education-related Federal activities and funding opportunities
3. Track progress and update the implementation plan

The White House's vision for STEM education was laid out in *Charting a Course for Success: America's Strategy for STEM Education* (Committee on STEM Education, 2018). The vision of the *Federal STEM Education Strategic Plan* is that:

All Americans will have lifelong access to high-quality STEM education and the United States will be the global leader in STEM literacy, innovation, and employment.

The Progress Report on the Federal Implementation of the STEM Education Strategic Plan (Office of Science and Technology Policy (OSTP), 2020) restates the three goals supporting this vision:

1. *Build Strong Foundations for STEM Literacy* by ensuring that every American has the opportunity to master basic STEM concepts, including computational thinking, and becoming digitally literate. A STEM-literate public will be better equipped to handle rapid technological change and will be better prepared to participate in civil society.
2. *Increase Diversity, Equity, and Inclusion in STEM* and provide all Americans with lifelong access to high-quality STEM education, especially those historically underserved and underrepresented in STEM fields and employment. The full benefits of the Nation's STEM enterprise will not be realised until this goal is achieved. Diversity includes geographic, race, ethnicity, gender, socioeconomic status, veteran status, parental educational attainment, disability status, learning challenges, and other social identities.
3. *Prepare the STEM Workforce for the Future* for both college-educated STEM practitioners and those working in skilled trades that do not require a four-year degree by creating authentic learning experiences that encourage and prepare learners to pursue STEM careers. A diverse talent pool of STEM-literate Americans prepared for the jobs of the future will be essential for maintaining the national innovation base that supports key sectors of the economy and for making the scientific discoveries creating the technologies of the future.

Four pathways for implementing the goals are also discussed in this report (Office of Science and Technology Policy (OSTP), 2020):

1. *Partnerships:* Develop and enrich strategic partnerships
2. *Convergence:* Engage students where disciplines converge
3. *Computational:* Build computational literacy
4. *Transparency:* Operate with transparency and accountability

The *Partnership Pathway* focuses on strengthening existing relationships and developing new connections between educational institutions, employers, and their communities. Partnerships can build STEM ecosystems to broaden and enrich education, engage students in work-based learning, blend formal and informal learning, and integrate curriculum for core academic and applied technical curricula in preparation for higher education.

The *Convergence Pathway* seeks to make STEM learning more meaningful and inspiring to students by focusing on complex real-world problems and challenges that require initiative and creativity. *Convergence* includes engaging learners in *transdisciplinary activities* such as project-based learning, problem-based learning, science fairs, robotics clubs, invention challenges, or gaming workshops. This pathway also calls for innovative instruction in mathematics, which frequently is a barrier to persistence in STEM, and advancing innovation and entrepreneurship education.

The *Computational Pathway* recognises how thoroughly digital devices and the internet have transformed society and adopts strategies that empower learners to take maximum advantage of this change. Computational thinking is central to this pathway, which includes solving complex problems with data and using computing devices effectively. This pathway promotes the use of digital platforms for teaching to provide anywhere/anytime learning that is customised for individual instruction, including the use of simulation-based or virtual reality experiences.

# FEDERAL STEM EDUCATION PARTNERS

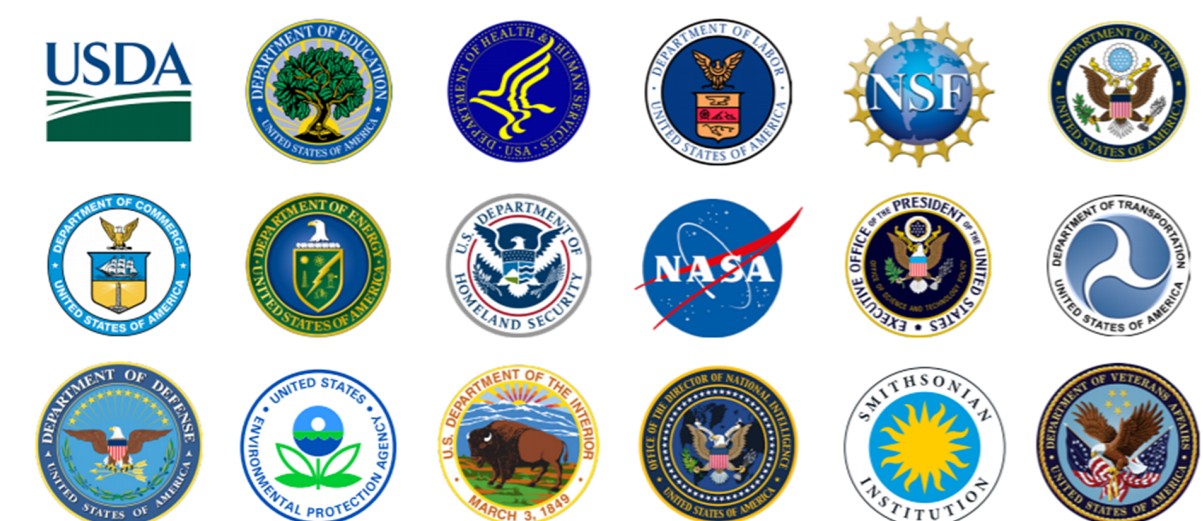

**Figure 1.** The Federal Coordination in STEM Education Subcommittee agency partners.

The *Transparency Pathway* commits the Government to open, evidence-based practices and decision-making in STEM programs, investments, and activities.

The Interagency Working Groups (IWGs) support FC-STEM in implementing the Strategic Plan (Office of Science and Technology Policy (OSTP), 2020). The FC STEM Strategic Partnerships IWG focuses on fostering STEM ecosystems that unite communities. The group's first objective is to establish additional connections between Federal STEM professionals and Federal facilities that support local and regional STEM ecosystems by providing opportunities for mentorship, educator professional development, curriculum material development, and other community engagement activities. The group's second objective is to expand the availability of high-quality, paid internships within Federal agencies and ensure that mentors are trained to provide effective educational experiences. The 18 Federal Agencies in Figure 1 are partners in this initiative.

The FC STEM Strategic Partnership IWG explores actions these Federal agencies can take to build STEM education ecosystems and promote work-based learning. STEM ecosystems consist of multi-sector partners united by a common vision of creating accessible, inclusive STEM learning opportunities that increase STEM literacy and expose learners to a variety of STEM career paths, as well as supporting student transitions from pre-K to STEM careers. Work-based learning experiences offer powerful, relevant ways to ensure STEM learning is authentic and engaging and prepares learners for the modern STEM workforce.

Now that we have discussed the federal agency players and their roles, let's focus on the convergence pathway and transdisciplinary STEM education practices.

## 3. Convergence

The FC-STEM Convergence IWG (Office of Science and Technology Policy (OSTP), 2020) has the following objectives:

1. *Transdisciplinary STEM Education Practices*: research, development, and dissemination of effective practices and federally funded programs.

2. *Transdisciplinary STEM Learner Support*: internships, fellowships, scholarships, and other training opportunities supporting transdisciplinary STEM learning.

Transdisciplinary teaching requires the integration of mathematics and statistics in STEM classes through meaningful and applied contexts. STEM educators will need professional development to enable them to reflect on and incorporate transdisciplinary approaches such as engineering design, entrepreneurship, computational reasoning, science model-based reasoning, or quantitative reasoning into their teaching (Bartholomew & Strimel, 2018; Purzer & Quintana-Cifuentes, 2019).

NSF views *convergence* as the merging of ideas, approaches, and technologies from widely diverse fields of knowledge to stimulate innovation and discovery (National Research Council, n.d.). First, convergence requires a compelling real-world problem, one that arises from a deep scientific question or a pressing societal need (Wals et al., 2014).

Grand challenges exist in all areas of STEM and society, providing a compelling set of real-world problems. These include:

1. Grand Challenges Initiative https://grandchallenges.org/
2. United Nations Sustainable Development Goals https://www.un.org/sustainabledevelopment/blog/tag/global-challenge/
3. National Science Foundation 10 Big Ideas https://www.nsf.gov/news/special_reports/big_ideas/convergent.jsp

These grand challenges crossover all areas of STEM and some are truly transdisciplinary. For an example of convergence in plant biology see Bender (2008), who lists the role of plants in addressing grand challenges in biology:

1. How do cells work and how do they interface with the environment?
2. How do single cells develop into multi-cellular organisms?
3. How do genomes generate organismal robustness and diversity?
4. What is the molecular basis of evolution?

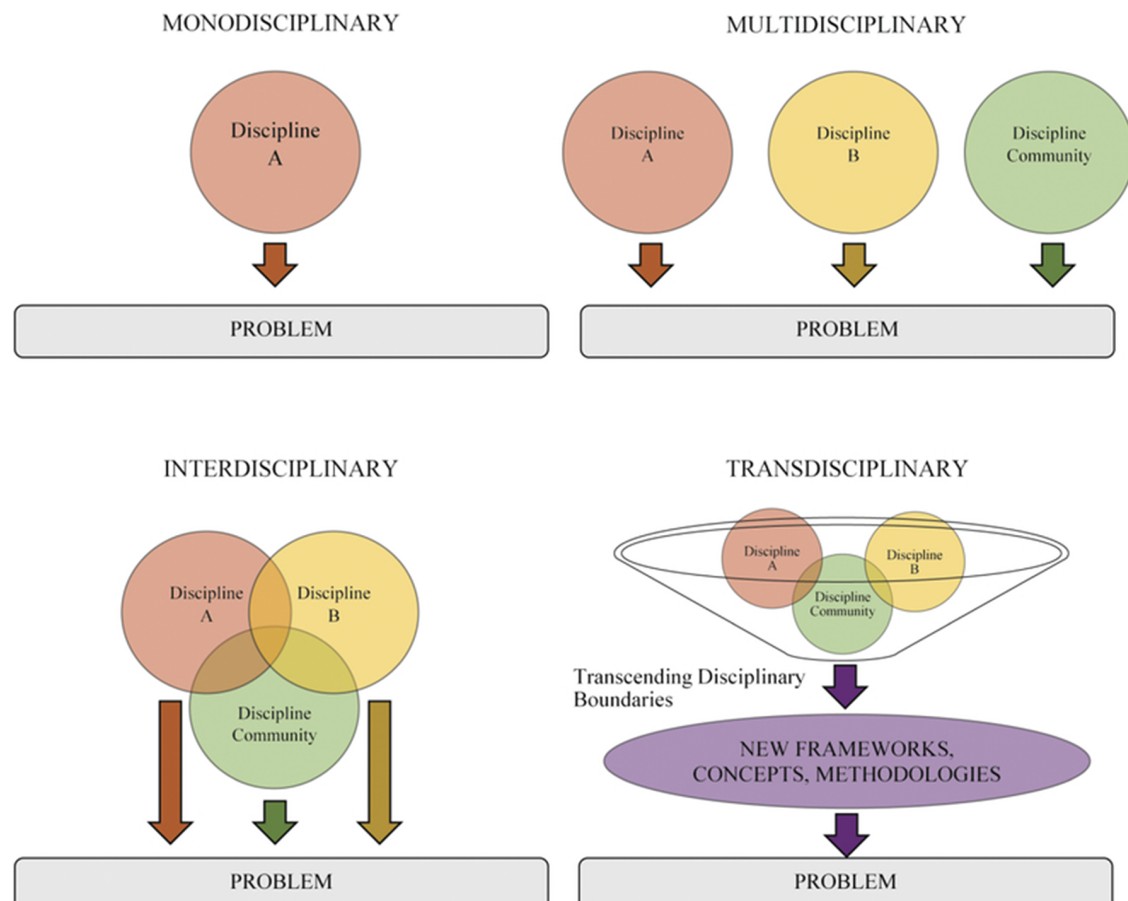

**Figure 2.** Levels of integration. Adapted from Heinzmann et al., 2019. Reprinted with permission.

5. How are biological systems integrated from molecules to ecosystems?
6. How can the environment be made sustainable for future generations?

Second convergence must be interdisciplinary, examining issues that lay at the overlap of STEM areas (Watson, 2017). These are problems that cannot be solved by examination in one domain but require STEM professionals from multiple disciplines to collaborate. Convergence requires integration across science, technology, engineering, and mathematics disciplines; however, STEM is taught using varying levels of integration (Vasquez et al., 2013). According to English (2016):

1. Disciplinary: concepts and skills are learned separately in each discipline
2. Multidisciplinary: concepts and skills are learned separately in each discipline but within a common theme
3. Interdisciplinary: closely linked concepts and skills are learned from two or more disciplines with the aim of deepening knowledge and skills
4. Transdisciplinary: knowledge and skills learned from two or more disciplines are applied to real-world problems and projects, thus helping to shape the learning experience

Think of the levels of integration across disciplines related to a plant biology problem (Figure 2).

A biologist could address the problem only from the discipline of biology. The biologist could form a multidisciplinary team with a mathematician and physicist, with each expert addressing the problem separately from their disciplinary viewpoint. The biologist could take an interdisciplinary approach where the biologist, mathematician and physicist view the problem through the lens of all three disciplines. But to reach the level of transdisciplinary STEM, the research team must be immersed in the social, ethical, economic, and/or political context of the real-world problem. Transdisciplinary approaches result in new frameworks, concepts, and methodologies that cross disciplinary boundaries.

The levels of integration can also be viewed through the elements of integration, perspective, team's goals and leadership (Figure 3).

## 4. Transdisciplinary

Teaching convergency requires moving to the transdisciplinary level of integration where there is deep integration of skills, disciplines, and knowledge to solve a challenging real-world problem. Transdisciplinary learning allows students to make authentic connections so they can construct their own meaning and transfer learning to real-world applications (Clark & Button, 2011). Research indicates that for transdisciplinary learning to be successful (a) the individual subject content foundations underlying the real-world problem must be mastered and (b) the learning must be intentional and explicitly outlined (National Research Council, 2014). While the first element of transdisciplinary learning is logical, when overdone, it can undermine student engagement in the

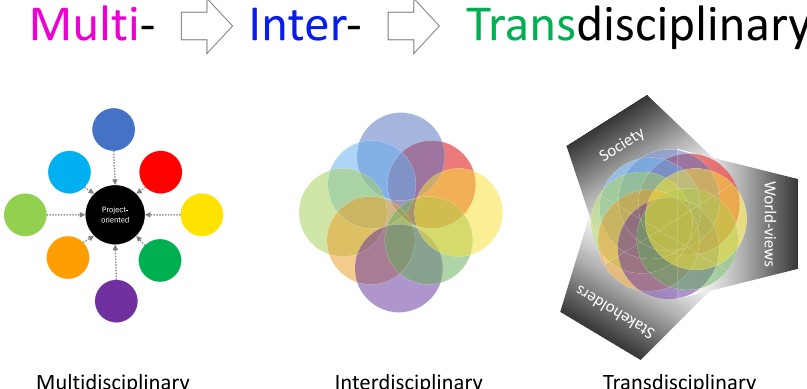

**Figure 3.** Levels of Integration Elements. Adapted from Awan, 2022. Reprinted with permission.

real-world problem. Just in time learning of a skill or concept that arises in solving a problem provides powerful motivation for the student. The student now has a reason for mastering the skill or concept. As an analogy, we do not ask a person to know all the elements of how a phone works before letting them use a phone. The user learns by discovery and invests the effort in making the discovery because they are motivated to use the phone. In real-world problem solving, STEM professionals often do not possess all the building blocks that will be needed, they attack the problem, circle back when needed to get the next block, then proceed. Creativity and problem-solving are not linear processes.

The second element of transdisciplinary learning is intentionality and explicitness, but these raise similar concerns. Whiletransdisciplinary learning may not happen just because you engage a student in solving a real-world problem, overplanning and direction of the learning can remove the creative aspect of problem-solving. Indeed, if overdone it turns an open-ended exploration into an over-guided tourist expedition where all the discovery is done for the student. This is indicative of one of the barriers to integrating transdisciplinary learning into the classroom. Curricula are driven by required practices and concepts that students are expected to master. If the instructor specifically designs an activity to combine the desired practices and concepts, then the transdisciplinary nature of the problem may be lost. If the instructor selects a real-world problem based on the practices and concepts to be solicited to meet the curricular goals, then student engagement may be lost. While transdisciplinary learning can elicit mastery of practices and concepts, a central goal is to incorporate transdisciplinary approaches such as engineering design, entrepreneurship, computational reasoning, science model-based reasoning, or quantitative reasoning.

Several goals for transdisciplinary learning have been identified, including STEM literacy, twenty-first-century competencies, STEM workforce development, and increasing and diversifying the STEM pipeline. Transdisciplinary learning has the potential to develop future STEM professionals who can work across disciplines to solve some of the world's greatest challenges.

How does STEM research differ across levels of integration at the convergence of STEM disciplines? Multidisciplinary, interdisciplinary, and transdisciplinary research all include a problem-solving focus, engage multiple disciplines, require knowledge sharing between disciplines, and coordinate research. Interdisciplinary and transdisciplinary research go beyond multidisciplinary research in implementing an iterative research process, integrating research, and crossing epistemological barriers. Transdisciplinary research pushes beyond interdisciplinary research in synthesising disciplines and theory, involving stakeholders, addressing societal dimensions of the real-world problem, and implementing the results found as part of the process of mitigating it.

Collaboration among stakeholders is essential if convergence is going to be integrated into STEM classrooms. K-12 teachers, college faculty, education researchers, professional organisations, and Federal Agencies need to develop a coherent framework for convergence in the classroom and collaborate to develop curriculum, assessments, and dissemination plans that promote the importance of convergence for the future of STEM (Margot & Kettler, 2019).

## 5. Convergence and transdisciplinary learning in plant biology

Here we provide examples of convergence as it relates to plant biology. Convergence is an approach to problem-solving that cuts across disciplinary boundaries. The modern workplace and cutting-edge research are convergent. What does it look like in plant biology? Below is a plant biology example that takes us through the four levels of integration.

1. Disciplinary: students learn concepts and skills separately in isolation
2. Multidisciplinary: students learn concepts and skills separately in each discipline, but in relevance to a common theme
3. Interdisciplinary: students learn concepts and skills from two or more disciplines that are tightly linked to deepen knowledge and skills
4. Transdisciplinary: students apply knowledge and skills from two or more disciplines to undertake real-world problems with societal dimensions to make the learning experience authentic

## 5.1. Duckweed case study of morphology and life cycles

*Disciplinary*—Biologist teaches about the life cycle of duckweed (a small aquatic plant that reproduces quickly) and sexual reproduction, Mathematician teaches about exponential growth in abstraction ($y=ax^b$).

*Multidisciplinary*—Biologist teaches about nutrient limitation and physiological effects and suggests the impact on growth rates or reproductive output. Mathematician investigates the biology of the coefficients and how variation in these coefficients changes the shape of the curve.

*Interdisciplinary*—Biologist has students predict growth patterns in various scenarios—high, medium, and low nitrogen levels and compare these to provided figures. They introduce duckweed as an option to absorb excess nutrients in wastewater treatment. Mathematician supports this learning by using the same scenarios and connecting to the coefficients, mainly the exponent with students predicting the values and shape of the curve and maximising growth.

*Transdisciplinary*—Students establish multiple replicates of microcosms with different nutrient levels, replicating wastewater treatment facilities and grow duckweed, plotting population size over time. Students fit multiple curves to the data and relate the coefficients to the scenarios. Students conduct an experiment to collect their own data, analyze the data, and draw conclusions. Instructors and students collaborate with various stakeholders to consider human health and societal implications of aquaculture and wastewater treatment.

At a disciplinary level, a plant biology instructor may introduce duckweed (*Lemna* spp. and *Wolffia* spp.), a small aquatic plant. Duckweed commonly uses vegetative (asexual) reproduction, although it also sexually reproduces, providing an excellent case study of morphology and life cycles (DeBuhr, 1991). It is also an important aquatic plant for aquaculture and human consumption, feed for animals, and water treatment to remove excess nitrogen and phosphorus (Appenroth et al., 2018; Landesman et al., 2005; Mohedano et al., 2012). A mathematics instructor may introduce a geometric growth equation, $N_t = N_0 \times e^{r \times t}$ to describe population growth in abstraction. While the mathematics instructor may describe these with biological jargon (e.g., $N_0$ is the initial population size and $r$ is the per capita growth rate), the jargon is unlikely to be grounded in a manner that makes connections between the biological entities and the coefficients.

At a multidisciplinary level, the instructors and students begin to connect the two disciplines. The biology instructor may ask students to quantitatively predict the growth effects of nutrient limitation or stresses. With duckweed, this could take the form of the increase in the number of flowers or leaves (for vegetative reproduction) that result from nutrient additions or examination of genetic mechanisms for different growth (Michael et al., 2021). The instructor may couple these thought experiments with computational modeling on birth rates and subsequently on the per capita growth rate. The mathematics instructor would also begin to layer on the biological realism that underpins the growth equations. Students would learn that the per capita growth rate is dependent on the per capita birth and death rate so changes to these would change the shape of the expected growth curve. The time variable would be related to the time until reproduction and consider meaningful time steps as well as meaningful starting conditions (N0). Students could determine a mechanistic understanding of growth rates including trophic efficiency and protein production rates.

At the interdisciplinary level, students are learning quantitative biology and instructors are purposefully structuring the pedagogy to consider both disciplines simultaneously. Students are predicting growth patterns in various nutrient or temperature scenarios and comparing these to provided data on duckweed growth. This could

be particularly important in the case where duckweed is considered in wastewater treatment (Landesman et al., 2005). Wastewater treatment facilities have a high nutrient water that provides perfect growing conditions for duckweed and duckweed are easily removed after growth. Duckweed establishment in various treatment facilities can provide a purpose for exploration as students reason about the variation in birth and death rates as mechanisms behind the different curves. Students readily translate tabular data into graphical representations, looking for patterns in both numeric and graphical data. In countries with water quality concerns, duckweed may help in nitrogen and phosphorus removal, providing students with real case studies to link biology with mathematical reasoning. At the interdisciplinary level, students develop hypotheses that reflect biological and mathematical contributions. They describe experiments that could resolve these hypotheses and generate data that they would expect from experiments. With duckweed, for example, they may predict a 5% greater per capita growth rate in high nitrogen compared to low nitrogen nutrient solution but not reduce the nitrogen levels sufficiently to release back into the environment. As they consider the nitrogen use efficiency of duckweed in treatment facilities, they will be able to determine whether duckweed alone is sufficient to meet threshold limits set by governments.

At the transdisciplinary level, students put their experimental design into practice and consider the social implications of knowledge about duckweed growth. Instructors have set up the course to fully integrate biology and mathematics within a socially-relevant context. For example, students may consider the consumption of duckweed by humans as a major source of protein in Southeast Asian culture (Appenroth et al., 2018), or the use of duckweed as a feedstock for chickens, swine, ducks, and carp that will later be consumed by humans (Sońta et al., 2019). Students develop hypotheses and experimental designs, as with the interdisciplinary level, then enact these designs and make predictions about social impact. Teaching collaboratively would allow students to learn from stakeholders who understand the social contexts and consider the factors influencing decision-making in these contexts. Duckweed can double its biomass in less than 1 week and the rapid feedback to the student from design to implementation to results to modeling the data, allow revision and re-implementation that is essential to using quantitative modeling effectively in the classroom. The inherent variation and abundance of data allow for experimentation, with both the biology and mathematical modeling. Importantly, instructors can emphasise the connections between the disciplines as students mathematically determine the coefficient (per capita growth rate) and relate it to the biology of the duckweed and the experimental conditions. As the instructors combine mathematics and biology, they can constantly revisit the cultural context that will determine the value of the duckweed to the local groups. For example, students can quantify the water quality and fish growth effects of growing duckweed in rural aquacultures to reduce nitrogen and phosphorus (Sarkheil & Safari, 2020) or compare duckweed grown for feed to duckweed for biofuel production (Cui & Cheng, 2015).

## 6. Transdisciplinary STEM and the societal dimensions of real-world issues

Although convergency and transdisciplinary integration better positions STEM disciplines to contribute to the resolution of real-world problems, current conceptions of transdisciplinary STEM limit its potential for doing so (Zeidler, 2016). As indicated in the duckweed example above, real-world problems are socioscientific

in nature. That is, though these issues are informed by the STEM disciplines, and a convergence of the STEM disciplines is requisite to their resolution, real-world issues cannot be effectively resolved without also addressing the societal dimensions of these issues (Zeidler, 2014). For example, Owens et al. (2021) recently investigated undergraduates' reasoning about a real-world issue playing out in the Midwestern United States, which they called the Raccoon River Nitrates issue. In short, nitrate runoff entering streams from massive corn farming operations is poisoning the water for citizens downstream who rely on it for their drinking water. Because the consumption of nitrate-laden drinking water reduces the ability of blood to deliver oxygen to the body, citizens of cities and towns downstream, such as Des Moines, are spending millions to filter the nitrate from the water. Whereas the farmers have long been protected from having to account for their nitrate runoff by the Clean Water Act, residents of Des Moines are not happy to be paying to address a problem that they did not cause—something they find to be unfair. Undergraduates attempting to resolve the Raccoon River Nitrates issue necessarily integrated content and practice from STEM disciplines in their reasoning, which included biology, ecology, hydrology, earth science, and physiology. However, not surprisingly, researchers also found undergraduates' reasoning to be informed by disciplines that were not STEM derivatives, such as economics, politics, psychology and sociology, as well as morality, ethics and aspects of character and civic virtue. Thus, if hopes for transdisciplinarity to solve the world's most pressing issues are expected to materialise, resolutions must truly be transdisciplinary and NOT limited to STEM disciplines.

## 7. Conclusion

The consensus of research is that a convergence focus in the classroom engages learners in STEM and develops problem solvers who can critically reason from multiple STEM perspectives (Roco & Bainbridge, 2013; Schmitz & Nikoleyczik, 2009; Shanahan et al., 2020). Convergence experiences also make STEM learning more meaningful for students and contributes to the development of workforce readiness skills through real-world problem-solving (Sengupta et al., 2019). However, although the convergence pathway is a step in the right direction, the notion that transdisciplinary STEM can effectively resolve real-world problems without including disciplines, such as ethics, economics or politics is unrealistic. We assert that transdisciplinary activities that inspire students to focus on understanding and responding to real-world problems need to take into account the societal nature of these issues by expanding interdisciplinary to include dimensions outside of STEM—if students are to be prepared to effectively address the socioscientific dimensions of real-world problems.

## Acknowledgments

This material is based upon work supported while serving at the National Science Foundation. Any opinions, findings and conclusions or recommendations expressed in this material are those of the author(s) and do not necessarily reflect the views of the National Science Foundation.

**Financial support.** This research received no specific grant from any funding agency, commercial or not-for-profit sectors.

**Competing interest.** The authors declare none.

**Authorship contribution.** R.M. wrote the convergence and transdisciplinary overview. J.D. wrote the plant biology examples. D.O. provided input on the social and economic aspects of transdisciplinary learning.

**Data availability statement.** No data or coding was used in this article.

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
