## [Reviewer Report]

Dear authors,

Thanks for your submission “Science, technology, engineering, and mathematics education efforts for quantitative biology” to QPB. My apologies for the time it has taken for this response. This was in part due to difficulty engaging reviewers over the summer period.

The editors have discussed your paper and comments from peer review are attached. There are several major points which we encourage you to address in a resubmission. From the editorial perspective, the manuscript as it stands is rather more orientated towards general policy and not yet scientifically-grounded enough for inclusion in QPB. The editorial recommendation is to reduce the bulk of the general statements and replace with concrete case study/studies in extenso to explain how data were collected, analyzed, etc. with a plant focus. The existing duckweed case study is somewhat limited in scope and detail and an expanded version would be more appropriate for QPB.

The reviewer’s comments should all be addressed but I’d like to highlight one of their foci in particular, regarding “transdisciplinarity”. They highlight that an important part of the transition from inter- to trans- comes with the inclusion of a social and ethical lens, which is currently rather absent from the definitions and the case study. They have some good suggestions for how to expand the final part of the case study to fold in these broader (and cross-sector) perspectives. This would automatically help the above point about expansion of plant science case studies.

---

## [Reviewer Report]

This article was accepted for publication by your journal, but was delayed by White House release. I have provided a Graphical Abstract as requested. All prior reviewer comments were addressed in the attached version.

---

## [Reviewer Report]

Thanks again for your time revising this manuscript. I believe comments from both reviewers and me have been addressed and am happy to recommend that we proceed with this publication.

For the record, the authors have obtained permission from the original content creators for the content reproduced in this article.